# Participation in Community-Based Solid Waste Management in Nkulumane Suburb, Bulawayo, Zimbabwe

**Ndidzulafhi Innocent Sinthumule [1],\* and Sinqobile Helen Mkumbuzi [2]**

[1]  Department of Geography, Environmental Management and Energy Studies, University of Johannesburg, P.O. Box 524, Auckland Park 2006, South Africa

[2]  Department of Ecology and Resource Management, University of Venda, Private Bag X 5050, Thohoyandou 0950, South Africa; sinq.mkumbuzi@gmail.com

\*  Correspondence: isinthumule@uj.ac.za

**Abstract:** After years of conventional approaches to solid waste management (SWM), in 2009, Bulawayo City Council adopted a non-conventional approach in the form of community-based solid waste management (CBSWM). The success of a CBSWM depends on the participation of members of the public as well as private sector organisations. Yet there is no information documented about their involvement in such activities in the study area. This study provides an analysis of citizen knowledge, participation and their attitudes in SWM in Nkulumane suburb following implementation of a CBSWM project. Door-to-door surveys were undertaken in December 2017 and January 2018 during which interview-administered questionnaires were used to collect data from 375 randomly-selected households. Semi-structured interviews were also used to gather data from officials responsible for CBSWM. The study found that the CBSWM has not been successful in changing the waste disposal behaviour of citizens. It was also found that the community-based organisations (CBOs) have made no effort to implement alternative waste management practices of waste recycling and composting. Furthermore, lack of funds to improve waste infrastructure and infighting between the Environmental Management Agency (EMA) and the Bulawayo City Council have undermined the principles of CBSWM. The study concludes by suggesting strategies that could improve CBSWM in developing countries.

**Keywords:** solid waste management; community-based organisations; household waste; local communities; Nkulumane suburb; community participation

## 1. Introduction

Solid waste management (SWM) has become a major concern both to the natural environment and society, particularly in developing countries. The rapid growth in population and the movement of people from rural to urban areas in search of jobs and better living conditions [1] has increased the volume of solid waste generated in urban centres [2,3]. The situation is made worse particularly when rapid-urbanisation results in the springing up of informal settlements [2]. The ever-increasing amount of waste is creating challenges for municipalities in managing waste [1,4,5]. In addition, when developing countries attained independence from colonialism, they adopted the conventional system of managing waste [3,6]. That is, they embraced a system in which waste is managed by local authorities. Local government departments oversee conventional SWM centrally and the costs of waste removal are fixed by the government [3,7,8]. In other words, waste management is generally seen as the responsibility of the local authority alone. This approach does not foster co-operation

between the community and the local authorities; instead, 'we dump–they collect' is the general attitude that conventional SWM fosters among the occupants [3,9]. In other words, this approach has low involvement of local communities and leaves limited room for civilians to participate [10].

This conventional model of SWM adopted from the colonial era has proven to be ineffective, inefficient and unsustainable for developing countries [11,12]. For instance, waste management efficiency under conventional methods in most developing countries is hindered by financial constraints that local authorities encounter since they operate under meagre budgets [8,13,14]. This has led to low waste collection levels or where garbage collection operation does not occur at all [15–17]. In turn widespread illegal open dumping has been triggered, particularly in urban areas [3,18]. Open dumping is not only a nuisance, but also an environmental problem which puts the health of residents at risk because the open waste dumps are breeding ground for mosquitos, rats, houseflies, and other vectors of infectious diseases. Illegal waste dumps are also a source of odours, water pollution and smoke emissions [18–21].

It is argued that municipalities in developing nations face problems of managing solid waste because they continue to rely on a 'collect, transport and throw away' approach. Conventional waste management systems have been criticised for their 'one size fits all approach' because they do not account for the fact that each town or city and its neighbourhoods has unique waste management needs [22]. It is also discredited in that this approach lacks the ability of solving waste management problems because it allegedly transfers 'the problem' (which is waste) from the source of waste generation to waste disposal sites. Furthermore, the system is considered to be land-intensive as vast tracts of land are required to cater for waste dumps and landfills [23]. These limitations of a conventional SWM system form a compelling argument for a much more comprehensive approach to SWM. In a bid to address the limitations of conventional waste management systems, a community-based approach to waste management, also known as participatory SWM, has been initiated in several developing cities [8,20,24,25].

In Zimbabwe, the Environmental Management Act (CAP 20:27) remains the principal act for addressing environmental issues including waste management. The Act also established an Environmental Management Agency (EMA) that is responsible for environmental management in the country. Section 95 of Act states that; "Every local authority shall prepare an environmental action plan for the area under its jurisdiction in accordance with such directions as the Minister may give". In adhering to EMA (CAP 20:95), Bulawayo City Council—the governing local authority—developed a CBSWM scheme in 2009 with the intention of addressing SWM issues. CBSWM is a waste management system that involves the development of a close-knit relationship between local authorities, private sector and communities [21,26]. However, no information is available regarding the participation of the general public and CBOs in SWM in the study area. Furthermore, no information is available concerning the current state of CBSWM in Nkulumane suburb. This study aims to investigate citizen participation in, knowledge of and attitudes to SWM in Nkulumane suburb following the implementation of a CBSWM in 2009 by Bulawayo City Council. This study intends to answer four main research questions. First, why did Bulawayo City Council adopt CBSWM? Second, how much do citizens participate in the implementation of CBSWM? Third, what are the attitudes and behaviour of local communities towards CBSWM scheme? Fourth, what are the achievements and struggles of the CBSWM in the study area? This paper consists of four sections and begins by giving a review of literature on CBSWM in developing countries.

*Some Basic Facts about Community-Based Solid Waste Management*

CBSWM is a waste management system that recognises the community as the active role player in cleaning up their neighbourhoods and/or to earning an income from solid waste [20,25,27]. The CBSWM approach is deep-seated on the principle of Kurt Lewin that states that people are likely to modify their own behaviour when they participate in problem solving. Thus, CBSWM gives people control over their environment to participate, maintain and improve its aesthetic value [24]. The role of

local communities is to practice sanitary behaviour achieved by keeping households and surroundings clean and storing waste in a designated bin/container [8,17,24]. Other roles of local people include resource recovery actions and participation both in consultation and in administration/management of solid waste services [20,28].

Participation by local communities might involve separating waste at household level, handing over separated waste to the waste collector and composting of organic wastes in backyards [20,24,29]. The compost generated by the community serves as the organic fertilizer [27] and this reduces the amount of organic matter dumped in landfills [20]. Most importantly, participation by local communities in waste management have adequately converted household waste from burden to resources through separation at source [2,25,30]. It appears that the success of CBSWM depends on the participation of local communities. This study sets out to contribute to the literature on waste management by providing an analysis of citizen knowledge, participation and their attitudes in SWM in Nkulumane suburb.

In areas where there is poor participation by local communities, it is argued that it could be improved if incentives were given to community members [2,30–32]. Furthermore, in CBSWM, community members are expected to attend meetings, elect leaders and representatives who manage waste collection and to give feedback and queries to the local authority [29]. Whilst the role of local communities may look simple on paper, it is argued that, realistically, the role that the community plays in managing their own waste is subject to local context as the whole exercise depends on the availability of strong local leaders and a competent local authority [33].

Local communities are not the only role players in CBSWM projects; rather, CBOs and local authorities are also important stakeholders. In a bid to solve waste management crises in urban areas, local private firms/companies are empowered by local authorities through CBOs to manage waste in urban areas [12]. CBOs complement the gap left by municipal authorities by providing SWM services at lower costs. Such services include encouraging separation of waste at source, recycling, waste recovery, composting, waste collection, waste treatment and transfer to disposal site [3,29,32,33]. In many developing nations, this has helped to reduce the rate of indiscriminate waste dumping [5,6]. In addition to CBOs, there are local authorities that are viewed as initiators and facilitators [11]. This essentially promotes cooperation, collaboration and the working together as partners in waste management between communities, local authorities and the private sector [3,18,34]. It is reported that CBSWM has led to improvements in waste management particularly in residential areas [32,33]. In other developing nations, the implementation of CBSWM has encountered social problems [26,35,36] and perceptions of waste management being solely the responsibility of municipalities [37]. These challenges have made CBSWM unsuccessful. This study investigates the current state of CBSWM in Nkulumane suburb in Bulawayo, Zimbabwe.

## 2. Material and Methods

### 2.1. Study Area

This study area is Nkulumane suburb, which is located at 28 31′01″ E and 20 11′06″ S (Figure 1). Nkulumane is one of the 156 suburbs situated in Bulawayo—the second-largest city in Zimbabwe. The city of Bulawayo is in Matabeleland Province; however, it is now treated as a separate provincial area from Matabeleland [38]. Bulawayo is 439 km southwest of Harare, the capital city of Zimbabwe. About 80% of the population in Bulawayo live in the high-density suburbs whilst the remainder live in the city centre and low-density suburbs [38]. Nkulumane is categorised as a high-density suburb populated mostly by low-income earners. Most of the people in Nkulumane area are of the Ndebele ethnic and language group followed by the Shona and Kalanga speakers, respectively. Nkulumane suburb is estimated to have a population of 151,824 distributed across 12,652 households with each housing unit, thus having an average of 12 people [39]. The study area is under the management of the Bulawayo City Council, which is responsible for service delivery such as sewer treatment,

water services and waste management. However, over the past 20 years, service delivery has seriously declined in Bulawayo and other parts of the country due to economic collapse. Zimbabwe currently sits on a 95% unemployment rate [40] and as a result, there is also a high rate of unemployment in Nkulumane suburb and people rely on activities including farming, mining and the black market for sustenance. Others depend on remittances sent by family members in other countries.

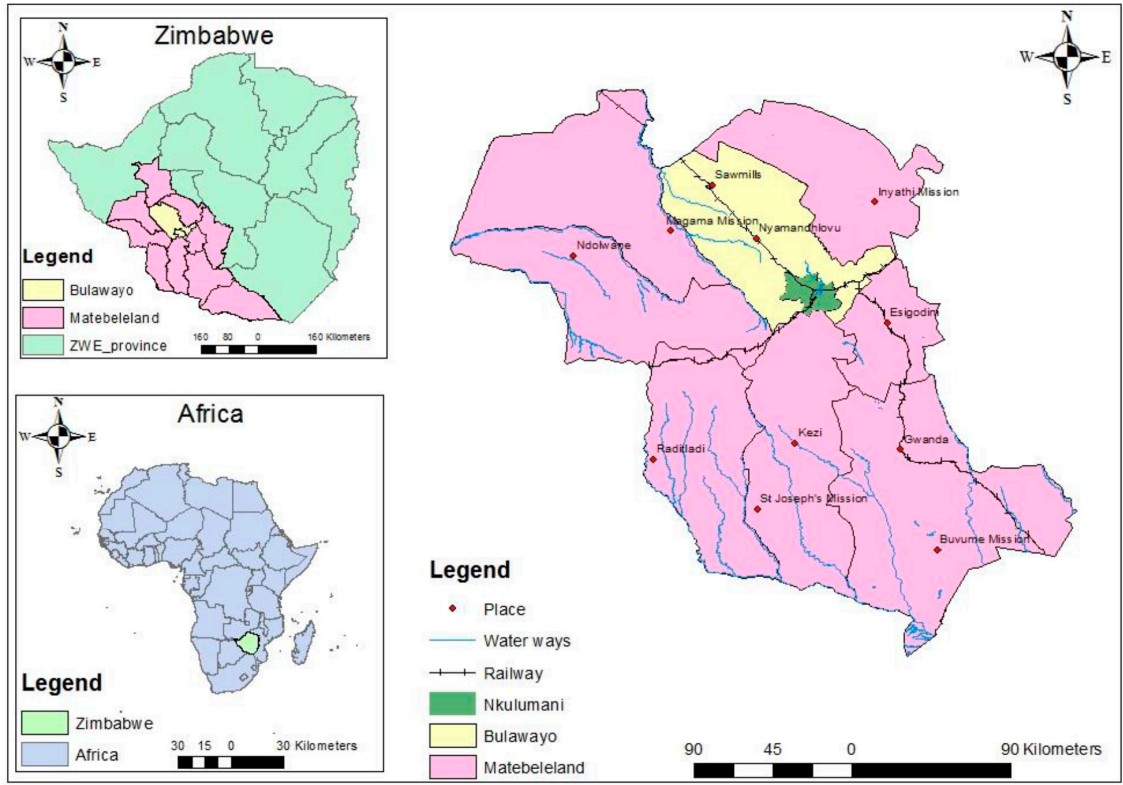

**Figure 1.** Location of the study area.

The study area is found at an altitude of 1360 m above sea level with an average annual temperature of 18.5 °C. The month of June has the lowest annual average temperature at 13.0 °C. The mean maximum temperature ranges from 20.5 °C in June to 29.2 °C in October. This means that October is the warmest month in the year falling at the peak of the dry season. Nkulumane area experiences three broad seasons, including a warm, wet period between November and April; a cool, dry winter from May to August; and a hot, dry period from August to early November [38]. The study area receives an average rainfall of 644 mm; this supports open woodland dominated by Terminalia and Combretum trees. Nkulumane area receives rainfall from October to April and the most rainfall falls in January with no rainfall received from June to August; July is the driest month in a year. Under the Köppen–Geiger classification system, Nkulumane suburb falls under a semi-arid climate (BSh) [38].

*2.2. Material and Methods*

The study relied on primary data collected in Nkulumane suburb during December 2017 and January 2018 following permission being granted by the Bulawayo City Council. The data was collected through: (i) semi-structured interviews; (ii) interview administered questionnaires; and (iii) observations. Semi-structured interviews were used to collect data from Bulawayo City Council officials who are responsible for CBSWM and the chairperson of CBOs that operate in Nkulumane suburb. The officials interviewed were from the Department of Amenities, Department of Environmental Health and Department of Public Housing in Nkulumane. Five personnel were purposefully selected and face-to-face interviews were conducted to find out the reasons for adopting

CBSWM and to determine the prospects and challenges facing the implementation of CBSWM in the study area (Appendix A). In addition, the Chairperson of CBOs was interviewed in order to understand their role, whether alternative waste management practices of waste recycling and composting were implemented to reduce waste and the challenges facing CBOs (Appendix B). The average duration of an interview was approximately 60 min. Data obtained from interviewing local authority officials and the Chairperson of CBOs was analysed using thematic content analysis. This is a qualitative analytic method for identifying, analysing and reporting patterns or themes within data [41].

Data on local communities were gathered using interview-administered questionnaires. This method was adopted in order to ensure that similar questions were asked of all respondents, thus avoiding bias and allowing for the calculation of statistical information such as percentages of questionnaire responses [42]. Importantly, interview questionnaires were administered because this allowed data to be collected through several means including closed-ended, open-ended and mixed questions (both open- and closed-ended questions in the same interview) [43–45]. This study employed mixed questionnaire interviews (closed-ended and open-ended questions) conducted face-to-face with the respondents. The open-ended questions were used to allow respondents to express themselves in their own words. This also helped the researcher to connect well with the informants.

Systematic sampling involving selecting samples based on a system of intervals [46] was used until 375 households had been covered. The questionnaires were designed to capture socio-economic and demographic characteristics, citizen knowledge, attitudes and their participation in CBSWM in Nkulumane suburb (Appendix C). Pre-testing of the questionnaire was undertaken on 20 people not part of Nkulumane suburb; this helped to shed light on the questions that were not clear [47]. All interviews were conducted by a master's student and three research assistants who were recruited from the local community in Nkulumane. The three research assistants were trained to administer the questionnaire to local communities. The household head (either male or female depending on who assumed responsibility for the household) or any adult member of the household above 21 years of age for selected households were interviewed using the questionnaires. The latter were translated to Shona and Kalanga (the local languages of the area) except where respondents were fluent in English. The total interview time ranged from 15–40 min. The participants were informed about the aim of the study and about the confidentiality of their responses. Respondents participated on the condition of anonymity and all respondents are referred to as anonymous in this research.

The collected data were recorded on a data sheet and later transcribed into English and tabulated in Microsoft Office Excel 2016 (Microsoft Corporation, Redmond, Washington, WA, USA). All analyses were performed using Statistical Package for Social Sciences (SPSS) version 25 for Windows (IBM SPSS Inc, Chicago, IL, USA). Chi-square ($\chi^2$) tests for goodness-of-fit were applied to find out whether or not the sampled respondents' socio-economic and demographic characteristics data were significantly different (at 5% significant level). This was important in order to compare the observed sample distribution with the expected probability distribution. Differences were considered to be significant at a 5% significance level. For open-ended questions, the researcher had to generate codes from the responses. The codes were generated by grouping similar responses from the questionnaires into one category. The codes were then registered into the SPSS software and a descriptive statistics tool was selected to analyse the codes that were generated from the questionnaires. This helped to generate frequencies up to 100% from the questionnaire responses.

## 3. Results and Discussion

### 3.1. Demography

Of the 375 respondents in this study, 54.4% ($n$ = 204) were women, with more women acting as respondents, if present, at most households ($\chi^2$ = 2.90, df = 1, $p$ = 0.08836). About 52% ($n$ = 195) of the respondents were between 21 and 29 years, while 25.1% ($n$ = 94) were between 30 and 39 years and 22.9% ($n$ = 86) were older than 40 years ($\chi^2$ = 59.06, df = 2, $p$ < 0.0001). Interviewees varied in

education level as follows: 13.9% ($n$ = 52) had at least secondary school qualifications (Ordinary (O) Level); 60% ($n$ = 225) had attained high school qualifications (Advanced (A) Level); and 26.1% ($n$ = 98) had tertiary education ($\chi^2$ = 124.04, df = 2, $p$ < 0.0001). Advanced level is done after Ordinary level by only those students who qualify. About 50.3% ($n$ = 189) of the respondents were self-employed, while 21.1% ($n$ = 79) were formally employed and the remaining 28.6% ($n$ = 107) were unemployed ($\chi^2$ = 52.29, df = 2, $p$ < 0.0001) (Table 1).

**Table 1.** Socio-economic profile of the respondents ($n$ = 375) in the study area.

| Characteristics | Class | % | $\chi^2$ |
|---|---|---|---|
| Gender | Male | 45.6 | - |
| | Female | 54.4 | $\chi^2$ = 2.90 |
| Age | 21–29 years | 52.0 | - |
| | 30–39 years | 25.1 | - |
| | >41 years | 22.9 | $\chi^2$ = 59.06 |
| Education | Secondary school | 13.9 | - |
| | High school | 60.0 | $\chi^2$ = 124.04 |
| | Tertiary | 26.1 | |
| Employment | Unemployed | 28.6 | - |
| | Self-employed | 50.3 | - |
| | Employed | 21.1 | $\chi^2$ = 52.29 |

### 3.2. Adoption of A CBSWM System

Up until 2009, Bulawayo City Council had a two-tier traditional system of waste removal. Waste was collected once a week within the Central Business District and in low density suburbs whilst in high-density areas waste was collected once in every 2–3 weeks. In other words, Bulawayo City Council was no longer providing adequate service within the Central Business District and the surrounding residential areas. Similar to the findings of the current study, Simon [3] as well as Kaseva and Mbuligwe [48] also reported that inadequate service delivery led to the adoption of CBSWM in Kinondoni Municipality, Dar es Salaam, Tanzania. The local authority claims that they collected waste less frequently in high density residential areas because their service fleet was overstretched, as made clear by one of the official: "As a city we require a certain number of vehicles to be able to manage waste efficiently, since we did not have enough and were overwhelmed, we acknowledged there were gaps in our service provision". The low frequency of waste collection particularly in high density areas prompted widespread illegal open dumping and backyard incineration. In a similar study in Kaduna Metropolis in Nigeria, insufficient number of waste trucks to cover their designated areas has led to households resorting to dumping their waste into public drains, in streams and along the roads [24]. Previous studies in Dar es Salaam, Tanzania [3] and Kenya (Nairobi, Mombasa, Kisumu, Nakuru and Eldoret towns) [49] also indicated that uncollected garbage by local authorities accumulated at roadsides, burnt by residents and disposed of in illegal dumping areas. The reasons for such poor performance in the study area were due to rapid population growth, urbanization and struggling economy. In a similar study in Kenya [49], rural-urban migration in search of better living conditions coupled with poor economic growth have been identified as the factors that led to problems and difficulties in managing waste in urban areas.

All these factors influenced Bulawayo City Council to decentralize SWM operations. Unlike in Lusaka, Zambia where the CBSWM project was funded by Danida (Danish International Development Agency) [26], in the study area, the project did not receive any donor funding. The CBSWM project in Nkulumane suburb was made possible by partnering with communities and the private sector to contribute in managing residential waste especially in high-density suburbs. Several previous studies have shown that participation and partnership with local communities is seen as an avenue towards sustainable waste management [7,8,17,32]. The idea of participation by local communities in this study

was made clear by one informant employed by the city council who reported that "In this new system of managing waste, we intended the community to rely less on the city council to clean after them but rather for them to be active in cleaning after themselves". Local communities are expected to at least separate inorganic from organic waste in their households. The communities are anticipated to dispose of organic wastes in their backyards to make compost. The idea is to reduce the volume of waste that is finally disposed into the landfill sites. In addition, local communities are expected to place waste bins along the roadside for collection by community truckers. Unlike in Putrajaya, Malaysia where local communities participate in solid waste segregation through alternative waste management practices of waste recycling programmes [8], there are no recycling programmes in Nkulumane suburb. In addition, unlike in Kaduna Metropolis in Nigeria [24] and Johannesburg in South Africa [50] where there are wheelbarrow or trolley boys (informal sector) who take recyclable material from households' waste, this is not the case in Nkulumane suburb. Singhirunnusorn et al. [51] argued that waste recycling from informal sector enhances the efficiency of recovering process, reduces the burden of disposal cost, and helps avoid the unnecessary and unhealthy disposal technologies. As a result, in the study area where informal sectors are not involved in waste recycling, local communities are unable to reduce garbage output and its impacts on the environment. Importantly, they are unable to earn an income from waste.

In a bid to solve suburban waste issues, communities are empowered by local authorities through CBOs to manage garbage in their own neighbourhoods [29]. In Bulawayo City Council, CBOs are subcontracted to handle waste. The tenders are given to those CBOs that have 5–7 roadworthy trucks. This approach is different from the one in Karachi, Pakistan where only the most educated are selected to serve as CBOs [15]. As in Dar es Salaam, Tanzania [3] and Jarkata, Indonesia [15], the role of CBOs in the study area includes giving awareness to communities on issues related CBSWM and organising clean-up campaigns. This is done by community-based street sweepers (CBSS) also known as waste pickers in India [52]. The CBSS are teams from local communities who are employed by CBOs on a contract (usually four-month) basis and earn up to US$200 (equivalent to R2300 per month). The role of CBSS is to load waste into trucks, ensure that the streets are clean and to monitor and report any form of illegal waste dumping and littering.

CBSS are employed on a rotational basis to ensure that each community member gets an opportunity for employment and they are selected with the help of ward councillors. Similar to the findings of this study, Simon also reported that CBOs in Dar es Salaam, Tanzania also collect waste from residences or households and are transported to the landfill site [3]. Furthermore, in Nairobi Kenya, a number of CBOs such as the Undugu Society and City Garbage Recyclers are involved in collecting recyclable materials (paper, metal scraps and plastics) that are sold to generate some income. Other groups are involved in composting of organic solid wastes (food wastes), which are sold to urban farmers or landscapers. In the town of Kisumu, Kenya one self-help group is now making mattresses from recycled polyethylene. These initiatives have successfully engaged large numbers of unemployed poor in gainful self-employment [49]. Similarly, in two Indian cities of Chennai and Hyderabad, Colon and Fawcett [33] also reported that CBOs implement alternative waste management practices of composting and recycling activities. The results indicate limited success of the schemes both in saving a significant fraction of the generated waste from dumping, and in rehabilitating the local poor [33]. In Nkulumane suburb, CBOs do not have these environmentally-friendly SWM practices of recycling and composting. This is owing to a lack of funding from government or donor agencies that can be used as incentives. This finding make CBOs in Nkulumane suburb, Zimbabwe to be different from other CBOs and this has negative implication in changing the waste disposal behaviour of citizens. The untreated household wastes in the study area are transported to waste collection stations where they are compacted and transported to the city's landfill site by city council vehicles. The waste that is collected by CBO truckers is put into bags and the local authority pays 23 cents (US$, equivalent to R2.30 per bag) for each bag delivered to the collection stations.

*3.3. Knowledge of a CBSWM by the Community*

Knowledge is a major barrier that can result in poor participation and poor separation of waste at source. This means that citizens who are better informed have a greater chance to participate in CBSWM than those who are not well informed [25]. Most respondents in the study area (68.8%, $n$ = 258) claim that they were aware of the existence of a community-based approach to SWM whereas the remaining (31.2%, $n$ = 117) reported that they had no knowledge of a CBSWM ($\chi^2$ = 53.01, df = 1, $p$ < 0.0001). However, those who were aware of the CBSWM further indicated that they were not adequately informed about the implications of combining wastes and the benefits of separation of waste at source. Previous research studies have indicated that awareness campaigns are key to a CBSWM and can have a positive impact on the community's attitudes towards waste management [5,14,53,54]. Awareness campaigns can be done through newspapers, internet, community meetings, radios and television. It is claimed that public knowledge and participation can be enhanced by combining the usage of all media [8]. However, according to the Bulawayo City Council, community members are given awareness on issues concerning a CBSWM by CBOs. The awareness campaigns are only done during community meetings. This raises an intriguing question—are awareness campaigns conducted only in community meeting sufficient? When informants were asked if they attend community meetings, (61%, $n$ = 229) said yes whereas the remaining (39%, $n$ = 146) admitted to absconding (Figure 2) from community meetings ($\chi^2$ = 18.37, df = 1, $p$ < 0.0001). It appears that attending community meetings is a low priority to some of the community members as made clear by one community member who argued that 'Since we are self-employed, time is money and we cannot afford to spend time doing anything else besides hustling for money'.

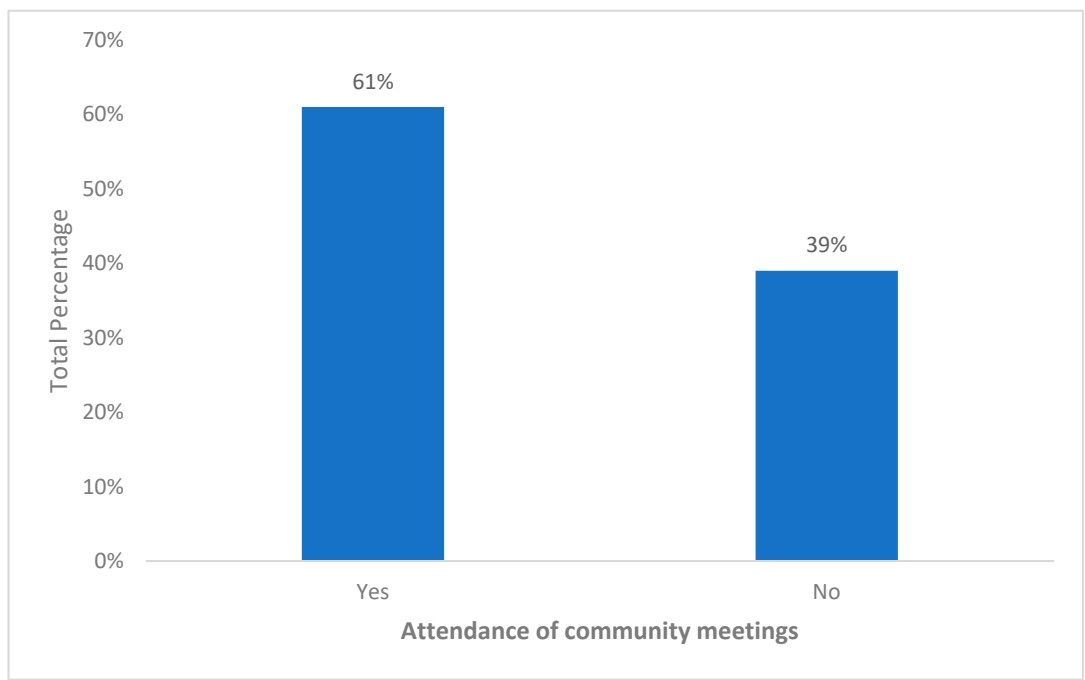

**Figure 2.** Attendance of community meetings in Nkulumane suburb.

*3.4. Knowledge of Reuse, Recycling and Reduction (3Rs)*

The terminologies of reduce, reuse, and recycle (3Rs) have become buzzwords when dealing with waste management. The success of these concepts depends on the participation of community and private sector which is the core of CBSWM. In this study, communities were asked if they were aware of the 3Rs in waste management. The study revealed that only 3.7% ($n$ = 14) of the populace had a good understanding of the 3Rs whereas 6.4% ($n$ = 24) had knowledge of only reuse and recycling. About 10.1% ($n$ = 38) of the respondents admitted that they had never heard of the 3Rs. About 10.4%

(*n* = 39) claimed they only knew what recycling was even though they were not practicing it and 10.1% (*n* = 38) of the population stated that of all the 3Rs they were only familiar with reduction of waste. The majority of the population (59.2%, *n* = 222) claimed that of all the 3Rs, they were only familiar with the reuse of waste (Figure 3).

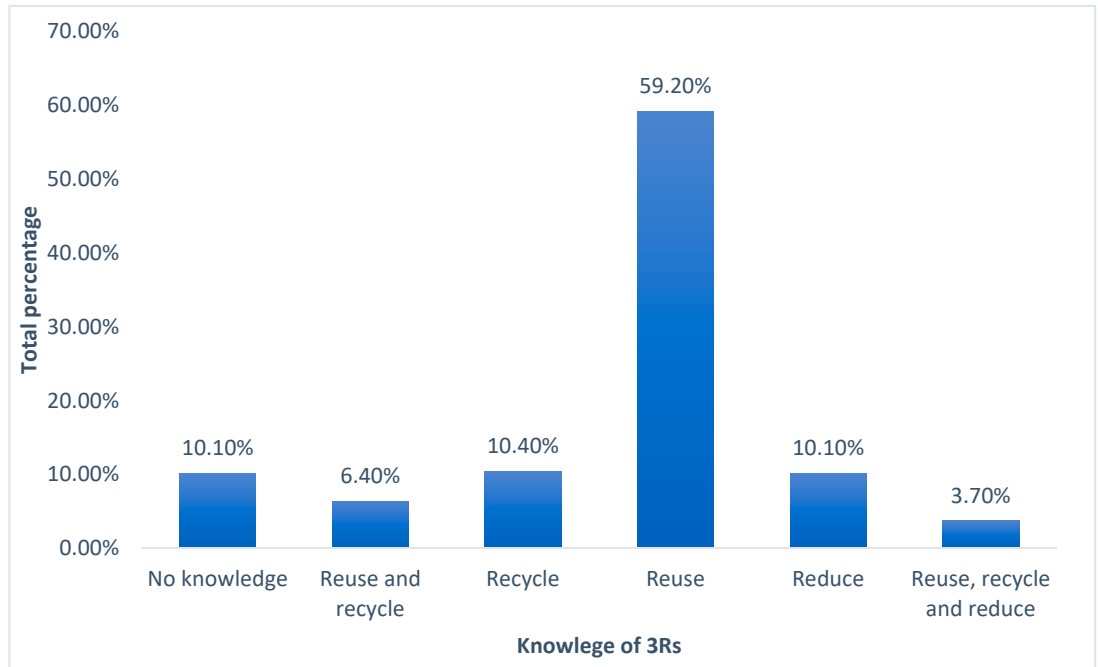

**Figure 3.** Knowledge of reuse, recycling and reduction (3Rs).

The participants further specified that of all the 3Rs, they were practicing reuse in their households. For instance, one informant indicated that she reuses plastic bottles to store water (Figure 4) since at times she experiences abrupt water cuts.

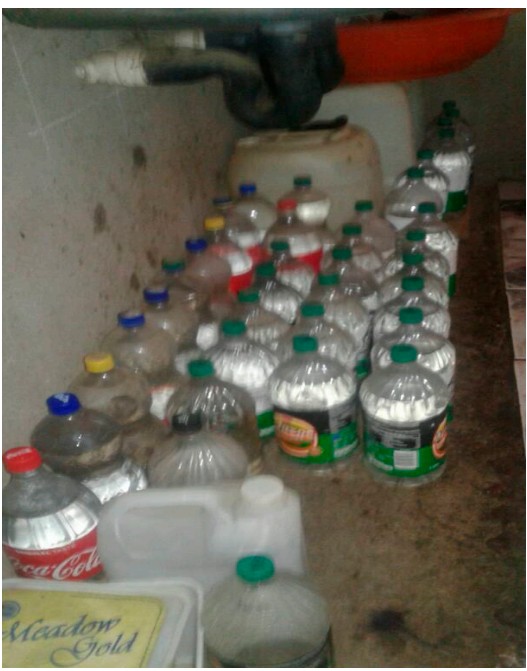

**Figure 4.** Plastic bottles are reused to store water by some community members.

Information on 3Rs behaviour in the household is important to understanding how the municipal waste problem might be resolved [49,55] by diverting most of the waste for reuse and recovery [56]. This encourages sustainable SWM [57] but, importantly, also reduces the costs of waste management [56]. In the study area, although the city council claim that CBOs conduct environmental awareness campaigns in community meetings on issues related to a CBSWM, only 3.7% understand all 3Rs. This means that local communities are not adequately informed about the 3Rs. Chu et al. [54] reported that lack of awareness is a major barrier in changing the waste disposal behaviour of citizens. This has negative implication on the success of CBSWM.

### 3.5. Community Participation, Behaviour and Attitudes towards CBSWM

One of the main reasons of moving from traditional ways of managing solid waste to CBSWM was to partner with community members to promote the working together as a team. Importantly, a CBSWM was established to encourage local communities to participate in waste management, particularly in residential areas [20,25]. When respondents asked how they handle their household waste, only 28.6% ($n$ = 107) of the informants indicated that they separate organic from inorganic waste whereas 71.4% ($n$ = 268) indicated that they throw all their waste in the waste bags and there is no systematic separation ($\chi^2$ = 68.76, df = 1, $p$ < 0.0001). These results demonstrate that although majority of people claim to be aware of CBSWM, they are not sufficiently informed about the importance of separating garbage at source. As a result, the attitudes and behaviour of a high proportion of respondents in the study area has not changed. These results are in line with Malik et al. [8] who reported that most people in Putrajaya Malaysia were opposed to separation of household waste because it was perceived as inconvenient. Similarly, a study carried out in Dhaka City Bangladesh [58] reported that majority of people (74%) did not separate the waste they generate because of lack of time, no economic incentives and no recycling facilities. As a result, this has not contributed to environmental improvement. The findings thus differ from those of Xiao et al. [25] who reported that the majority of respondents (53.5%) in their study based in Xiamen China always separate recyclable material from household waste. At the end, waste management in Xiamen has yielded some good results and has achieved a high level of citizen satisfaction. In all these case studies, separation and non-separation of household waste at source was not correlated with the level of education.

When asked how they handle waste when they are outdoors, 67.2% ($n$ = 252) of the respondents in the present study indicated that they throw litter in the bin when they are outdoors whereas about 32.8% ($n$ = 123) admitted to throwing litter anywhere convenient ($\chi^2$ = 44.38, df = 1, $p$ < 0.0001). Those who confessed to throwing waste anywhere indicated that they are forced to litter because the waste bins were few and in a dilapidated state, but importantly, the outdoor bins were unevenly distributed in the area. Those who stated they throw waste in dust bins indicated that they do not separate waste because there were no containers that allow them to do so. Metcalfe et al. [59] suggested that the provision of the required infrastructure is vital to encouraging separation of solid waste by local communities and may result in greater participation in SWM. About 20.27% ($n$ = 76) also indicated that they burn their waste at home despite the implementation of CBSWM in the area. The attitudes and behaviour of littering when communities are outdoors and burning of waste in backyards undermines the principles of CBSWM.

When respondents asked if they had ever dumped waste in undesignated places, 69.6% ($n$ = 261) indicated that they did not dump waste in undesignated places because they feared to be fined or punished. One community member recounted, "If those officials from EMA catch you dumping waste in undesignated places, you are in big trouble. They will order you to clean up the street as punishment, I will never risk embarrassing myself like that." About 16.27% ($n$ = 61) of the respondents admitted to dumping waste in undesignated places (Figure 5) and they reported doing it at night to avoid being caught by EMA monitors. The main reason given by respondents for dumping in undesignated area was that the bins in the households were full. The remaining 14.13% ($n$ = 53) of the respondents said that they had never dumped waste illegally because they believed that it is wrong ($\chi^2$ = 222.21, df = 2,

$p < 0.0001$). It is clear that in Nkulumane suburb, the majority avoid waste dumping in undesignated areas not because they perceive it to be wrong or bad for the environment. Rather, a conscious decision not to dump waste illegally is made because the communities fear "fines and punishment". This means that local communities have the wrong attitudes to waste management and their actions undermine the aims of CBSWM. A previous study in Kaduna Metropolis in Nigeria [24] also indicated that most respondents reported using the unofficial road side dumps for their household wastes despite the implementation of CBSWM. These results demonstrate that local communities still do not understand the benefits of implementing CBSWM in promoting their environment and health.

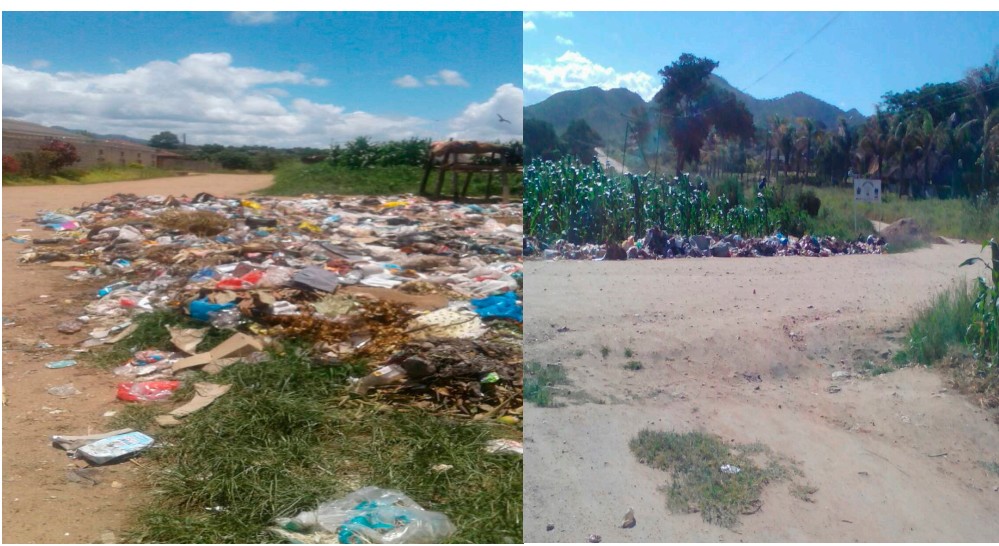

**Figure 5.** Illegal dumping sites in Nkulumane suburb.

The study also found that some community members do not appreciate the role that community-based street sweepers (CBSS) play in a CBSWM. This is evidenced by the negative attitude that some community members exhibit towards CBSS. It was revealed that there was a stigma associated with being a street sweeper. One informant recounted, "All sorts of names are thrown at us because we work with waste. It makes us uncomfortable and at times we are embarrassed to do our duties in the community streets because people look down upon us." It is common knowledge within the community that those who are selected to be CBSS are usually those who are economically vulnerable. This is in line with Colon [60] who argued that community waste collectors have a low social standing and community members do not respect them because they are viewed to be smelly and filthy people. This undermines the principles of CBSWM.

### 3.6. Achievements and Struggles of A CBSWM

Over the years, the few vehicles that Bulawayo City Council owned would run from door to door to collect waste in residential areas. This increased the rate of wear and tear of the waste fleet. In addition, the local authority spent a lot of money on fuel and incurred high expenses in servicing waste vehicles. Since CBOs are now responsible for transporting waste from the community to a central place using their own trucks, city council vehicles no longer break down as easily as in the past. Furthermore, the city council spends less in servicing their waste fleet since it now covers less mileage. The establishment of CBSWM has empowered community members, particularly the youth who are employed as waste loaders and street sweepers. Simon [3] and Wahab [61] also found that a move to a CBSWM helped to create jobs for the community. This allows community members to meet their financial obligations in their households. Importantly, community members who are contracted as CBSS are trained before operating. This allow them to gain knowledge on SWM. This increases community awareness on issues related to waste management in the area. CBSWM is task-based,

meaning that if one does not work they do not get paid and this promotes accountability. At the time of data collection, the local authority had financial constraints. As a result, they acknowledged that they did not have the necessary incentives that can be used to encourage the local communities to separate or recycle their waste. Rathi [30] argued that community participation can be boosted if incentives were given to community members. For instance, in Yala City in Southern Thailand, a package of new practices was introduced that was termed "garbage for eggs" in order to motivate the community to participate in waste management activities. Residents were encouraged to bring recyclable material in exchange for eggs [32]. An incentive-based source separation model was also used in the Nakhon Ratchasima metropolitan area in Northeastern Thailand [31] and Zhenhua Community located in Wudang District, Guiyang, Guizhou Province, in Southwest China [62]. The results showed that in all these case studies, this approach increased the waste separation in residential areas which improved community SWM. However, in the case of Nkulumane suburb in Zimbabwe, there are no incentives or rewards and the scheme has not yet changed the waste disposal behaviour of citizens.

It was found that local authorities do not have money to buy the containers that can enable separation of waste, as explained by an official from the city council, "To assist community members to separate waste at source, we need money to source the proper receptacles that can enable separation of waste. At the moment we don't have money to do so. As a local authority, we are not yet there." In addition, there are also institutional challenges. The study established that there is infighting between the EMA and the city council and the conflicts hindered the city council from meeting their obligations. It was revealed that if EMA comes across mass waste dumping and pollution in the community, they fine the city council. However, when EMA receives money for fines, they do not invest the money back into rehabilitating the environment. It is, therefore, significant that municipal authorities, waste management officials and policymakers should take into account these identified challenges while planning, implementing and/or redesigning CBSWM programmes.

## 4. Conclusions and Recommendations

The findings of this study show that the implementation of CBSWM has not yet reduced inappropriate waste disposal behaviour on local communities and in turn has not yet contributed to environmental improvement in the study area. In addition, the CBOs that have been subcontracted to handle waste have made no effort to implement alternative waste management practices of waste recycling and composting. To support the aims and objectives of CBSWM, there is a need to develop more environmental-friendly SWM (waste composting and recycling) programmes due to availability of both raw materials for these two practices. These industries hold significant potential to provide income for majority of poor people in the study area. It appears that the local people in the study area have still not yet completely understood the importance and benefits of implementing CBSWM in promoting their environment and health. As a result, there is a great need to increase citizens' awareness and responsibility toward solid waste source separation by using educational programs (e.g., environmental health campaigns) and public education in radios, television, newspapers, leaflet drop and others.

It is also important that local authorities should provide the required municipal solid waste infrastructure (e.g., proper waste containers) to allow citizens to separate waste at source. This is important because if the infrastructure to collect waste is not available, there is little that citizens can do. Where the funds are not available to buy the required infrastructure, there is a need to apply from donors who are interested in waste management programmes. In addition, there is a need to encourage separation of waste at source by giving local communities incentives. In some cases, local authorities should run competitions and give awards to the cleanest wards within a municipality. The suggestions presented in this article have the prospective to improve CBSWM project not only in Nkulumane suburb in Zimbabwe, but also in other developing economies where sustainable waste management practices are yet to meet a critical mass of success.

**Author Contributions:** Conceptualization: N.I.S.; methodology: N.I.S. and S.H.M.; software: N.I.S. and S.H.M.; validation: N.I.S. and S.H.M.; formal analysis: S.H.M. and N.I.S.; investigation: S.H.M.; resources: S.H.M. and N.I.S.; data curation: N.I.S. and S.H.M.; writing—original draft preparation: N.I.S.; writing—review and editing: N.I.S.; visualization: N.I.S.; supervision: N.I.S.; project administration: N.I.S. and S.H.M.

**Funding:** This research received no external funding.

**Acknowledgments:** The authors gratefully acknowledge all the respondents in Nkulumane suburb in Bulawayo, Zimbabwe who voluntarily participated in this study. We are very grateful to the anonymous reviewers whose comments and remarks contributed to the improvement of the quality of the paper.

**Conflicts of Interest:** The author(s) declare there are no potential conflicts of interest with respect to the research, authorship, and/or publication of this article.

## Appendix A. Interview Guide for City Council Officials Who are Responsible for CBSWM and the Chairperson of CBOs

1. What motivated the move from conventional waste management to CBSWM?
2. What measures have you put in place to ensure;

   - Proper storage of waste
   - Waste is separated at source
   - Waste is recycled, reused and reduced
   - Zero littering
   - No illegal dumping of waste

3. How do you ensure that the communities are given awareness on health, hygiene and the environment and how often does this happen?
4. What is the criteria for selecting CBOs and what are their role in SWM?
5. What are the challenges associated with initiating and spearheading the CBSWM scheme?
6. What have been your achievements as the local authority?

## Appendix B. Interview Guide for the Chairperson of CBOs

1. What are the criteria used to select CBOs?
2. What are the role of CBOs?
3. Do you implement alternative waste management practices of waste recycling and composting and why?
4. What has been achieved so far?
5. What are the challenges facing CBOs?

## Appendix C. Questionnaire Guide for Local Communities

1. What are the characteristics of your household waste?
2. How do you handle household waste and why?
3. Do you burn household waste?
4. How is your household waste collected? By who and how frequent?
5. How do you handle solid waste when you are outdoor and why?
6. What is your general understanding of CBSWM?
7. Have you been given any awareness on health, hygiene and the environment? If yes how frequent?
8. Who is responsible for giving the community awareness?
9. Do you know the importance of CBSWM?
10. Do you attend all the community meetings, if not why?
11. What do you understand about recycling, reuse and reduction of waste?
12. Do you recycle and reuse your waste?

13.　What kind of waste do you reuse and recycle?

14.　Have you ever dumped any of your waste in undesignated waste dumps? Why?

15.　What have been the benefits of being part of a CBSWM initiative?

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
