# Peer review of "Participation in Community-Based Solid Waste Management in Nkulumane Suburb, Bulawayo, Zimbabwe"

_resources, doi:10.3390/resources8010030_

Round 1
Reviewer 1 Report
Authors have a good job in presenting the methods, facts and the analysis. Well written and well done. Few suggestions:
- In the title the first phrase"people as participants" is redundant as the next phrase begins with "community based..."
- In the abstract last line should be changed to ".....by suggesting strategies that could improve CBSWM.
- line 159: I believe zero is mistakenly typed as "o" (not 0)
- Table 1: it could help readers if O-level and A-level is defined or replaced with middle school and high school as defined in the text
- In few places the "refuse"has been used to indicate waste. It is better to be consistent with the usage of waste as refuse is some times defined as waste minus recyclables.
- line 236: bridges is not clear. Did you mean rivers?
- Section 3.5 bring example from Asia (Malaysia and China) about people participation. This could have been a nice opportunity to see if there was a correlation with the level of education.
- line 373: check grammar
- Conclusions: since some recommendations are also made, it is advisable to change the title to conclusions and recommendations.
Author Response
Reviewer 1
Comments and Suggestions for Authors
Authors have a good job in presenting the methods, facts and the analysis. Well written and well done. Few suggestions:
Comment: In the title the first phrase "people as participants" is redundant as the next phrase begins with "community based..."
Response: The title has been modified in page 1.
Comment: In the abstract last line should be changed to ".....by suggesting strategies that could improve CBSWM.
Response: Last sentence in the abstract has been changed as suggested.
Comment: line 159: I believe zero is mistakenly typed as "o" (not 0)
Response: This has been corrected in page 4.
Comment: Table 1: it could help readers if O-level and A-level is defined or replaced with middle school and high school as defined in the text
Response: O-level and A-level has been replaced with secondary school and high school.
Comment: In few places the "refuse"has been used to indicate waste. It is better to be consistent with the usage of waste as refuse is some times defined as waste minus recyclables.
Response: The use of the word waste has been used throughout the article.
Comment: line 236: bridges is not clear. Did you mean rivers?
Response: The word bridges has been removed in page 6 so that the sentence read well.
Comment: Section 3.5 bring example from Asia (Malaysia and China) about people participation. This could have been a nice opportunity to see if there was a correlation with the level of education.
Response: This has been clarified in page 10.
Comment: line 373: check grammar
Response: grammar improved in page 10.
Comment: Conclusions: since some recommendations are also made, it is advisable to change the title to conclusions and recommendations.
Response: The words and recommendation are added
Reviewer 2 Report
The success of solid waste management and the way to a circular economy has to involve citizens. It is very important to study cases in this field to help us reflect on the models to be adopted.
So, the theme is very important and the paper is well written and well organized, however, it can be improved if the authors:
Present the content of the survey
Present the results graphically
Present more recent bibliography
line 145 - Are you sure that the unemployment rate is 90%? I have been checking and, in statistics, the highest unemployment rate in the world is far from the one that you presented.
Author Response
Reviewer 2
Comments and Suggestions for Authors
The success of solid waste management and the way to a circular economy has to involve citizens. It is very important to study cases in this field to help us reflect on the models to be adopted.
So, the theme is very important and the paper is well written and well organized, however, it can be improved if the authors:
Comment: Present the content of the survey
Response: The content of the survey is now presented as appendix 1, 2 and 3 in page 16.
Comment: Present the results graphically
Response: Graphic presentation is now included in page 8 and 9.
Comment: Present more recent bibliography
Response: This article has been written using recent literature with some published in 2018 and 2017. Nonetheless, more recent bibliography have been added.
Comment: line 145 - Are you sure that the unemployment rate is 90%? I have been checking and, in statistics, the highest unemployment rate in the world is far from the one that you presented.
Response: The facts have been verified and the sentence
Reviewer 3 Report
Introduction:
Page 2, lines 85-87: “This is followed by the methods employed in collecting and analysing data and the description of the study area. The results and discussions are presented in Section 3 while Section 4 sets out conclusions to the study.” This sentences are unnecessary.
Page 3, line 128 -129: “In the next section the study area and the methods employed to collect and analyse the data are described”. This information is needless.
Material and methods:
The lack of example of questionnaire.
Results and Discussion
In my opinion, a graphic presentation of waste management systems will make more attractive this part of article.
References
In my opinion, it is advisable not to cite references before 2005 year due to their distant year of publication.
Author Response
Reviewer 3
Comments and Suggestions for Authors
Introduction:
Comment: Page 2, lines 85-87: “This is followed by the methods employed in collecting and analysing data and the description of the study area. The results and discussions are presented in Section 3 while Section 4 sets out conclusions to the study.” This sentences are unnecessary.
Response: Sentences removed
Comment: Page 3, line 128 -129: “In the next section the study area and the methods employed to collect and analyse the data are described”. This information is needless.
Response: Sentences removed
Material and methods:
Comment: The lack of example of questionnaire.
Response: The sample of interview questions and a questionnaire is now included as appendix 1, 2 and 3.
Results and Discussion
Comment: In my opinion, a graphic presentation of waste management systems will make more attractive this part of article.
Response: Graphic presentation is now included in page 8 and 9.
References
Comment: In my opinion, it is advisable not to cite references before 2005 year due to their distant year of publication.
Response: There are only 6 references before 2005 and I find them adding value to this article.